# The Association between Metformin Use and Risk of Developing Severe Dementia among AD Patients with Type 2 Diabetes

**DOI:** 10.3390/biomedicines11112935

**Published:** 2023-10-30

**Authors:** Ying Xue, Xiangqun Xie

**Affiliations:** 1Department of Pharmacy and Therapeutics, School of Pharmacy, University of Pittsburgh, Pittsburgh, PA 15261, USA; 2Department of Pharmaceutical Sciences and Computational Chemical Genomics Screening Center, Pharmacometrics & System Pharmacology (PSP) PharmacoAnalytics, School of Pharmacy, Pittsburgh, PA 15261, USA; 3National Center of Excellence for Computational Drug Abuse Research, University of Pittsburgh, Pittsburgh, PA 15261, USA

**Keywords:** diabetes, Alzheimer’s disease, severe dementia, metformin, MMSE, APOE

## Abstract

This study explores the potential impact of metformin on the development of severe dementia in individuals with Alzheimer’s disease (AD) and type 2 diabetes mellitus (T2DM). With an emerging interest in the role of the *APOE* genotype in mediating metformin’s effects on cognitive decline in AD patients, we sought to investigate whether metformin usage is associated with a reduced risk of severe dementia. Using data from the National Alzheimer’s Coordinating Center (NACC) database (2005–2021), we identified 1306 participants with both AD and T2DM on diabetes medications. These individuals were categorized based on metformin usage, and a propensity score-matched cohort of 1042 participants was analyzed. Over an average follow-up of 3.6 years, 93 cases of severe dementia were observed. A Kaplan–Meier analysis revealed that metformin users and non-users had similar probabilities of remaining severe dementia-free (log-rank *p* = 0.56). Cox proportional hazards models adjusted for covariates showed no significant association between metformin usage and a lower risk of severe dementia (HR, 0.96; 95% CI, 0.63–1.46; *p* = 0.85). A subgroup analysis based on *APOE ε4* carrier status demonstrated consistent results, with metformin use not correlating with a reduced severe dementia risk. In conclusion, our findings from a substantial cohort of AD and T2DM patients suggest that metformin usage is not significantly associated with a decreased risk of severe dementia. This observation persists across *APOE ε4* carriers and non-carriers, indicating a lack of genotype-mediated effect.

## 1. Introduction

Alzheimer’s disease (AD) is a progressive neurodegenerative disorder characterized by memory loss, cognitive decline, and the impairment of daily functioning. It is the most common cause of dementia among older adults, affecting 47 million people worldwide [1]. Diabetes mellitus (DM) is a metabolic disorder marked by persistent hyperglycemia, stemming from various mechanisms that result in insufficient insulin production by the pancreas or the body’s cells becoming unresponsive to the hormone’s regulatory effects [2]. Type 2 diabetes mellitus (T2DM) accounts for over 90% of all diabetes cases in individuals [3]. Increasing evidence has suggested that diabetes might elevate an individual’s susceptibility to AD. A population-based study revealed that incidence of AD was approximately 2 in 1000 individuals for those without diabetes. However, for individuals with diabetes, the rate notably increased to 6.25 cases per 1000 people [4].

Numerous studies have shown that individuals with T2DM often exhibit worse cognitive performance when compared to their non-diabetic counterparts [5,6,7]. These findings indicate a potential association between T2DM and cognitive decline, which could be relevant to the development of AD. The association between AD and T2DM goes beyond the observation of cognitive impairments. Shared pathophysiological mechanisms underlie both conditions, such as insulin resistance and dysregulation of glucose metabolism [8,9,10]. Insulin resistance is a hallmark feature of T2DM, and it also affects the brain, disrupting insulin signaling and glucose uptake in neurons. This leads to reduced brain energy metabolism, which may contribute to the neuronal dysfunction and cognitive decline observed in both T2DM and AD.

Given the link between AD and T2DM, there is an ever-growing interest within the scientific community to delve deeper into understanding the potential impact of antidiabetic medications on AD outcomes and cognitive performance in patients with AD [11]. This intrigue is driven by emerging evidence suggesting that specific antidiabetic agents, such as metformin, might hold promise in mitigating the risk of AD development [12,13]. Notably, metformin, a widely used antidiabetic medication, has demonstrated its capability to enhance insulin sensitivity and reduce inflammation, properties that could potentially confer substantial benefits for overall brain health. However, it is important to acknowledge that the bulk of prior research has predominantly centered around examining the associations between metformin usage and dementia risk in individuals with T2DM who exhibit normal cognitive function [14,15,16]. Regrettably, few studies have evaluated the relationship between metformin use and cognitive outcomes among people with diabetes who also have a diagnosis of AD. This knowledge gap underscores the significance of further investigation into the association between antidiabetic drug use and cognitive progression among AD patients.

Apolipoprotein E (APOE) is a critical protein in the human body involved in lipid transport and supporting brain injury repair. The *APOE* gene polymorphisms are the primary genetic determinants of AD risk, with individuals carrying the *ε4* allele being at a higher risk compared to those with the more common *ε3* allele, while the *ε2* allele is associated with a reduced risk [17,18]. Moreover, the presence of the *APOE ε4* allele is also linked to an increased risk of cerebral amyloid angiopathy and an age-related cognitive decline during normal aging. A recent study suggested an association between metformin use and faster decline in delayed memory among the *APOE ε4* carriers [19]. The findings point to more research being needed to investigate whether the therapeutic effects of metformin are influenced by the *APOE ε4* genotype.

Considering the scarce research available on the impact of metformin on cognitive progression among AD patients and the necessity to explore potential interactions with the *APOE ε4* genotype, this study aims to investigate whether use of metformin is associated with a decreased risk of developing severe dementia in patients with AD and T2DM. Furthermore, this study seeks to examine whether the association between metformin use and cognitive outcomes is modified by the *APOE ε4* genotype.

## 2. Methods

### 2.1. Data Source

The National Alzheimer’s Coordinating Center (NACC) was established in 1999 by the National Institute on Aging (NIA). The NACC database consists of data from more than 42 Alzheimer’s Disease Research Centers (ADRCs) throughout the United States. Data from different ADRCs are structured in a standardized manner to form a Uniform Data Set (UDS). Recruitment, participant evaluation, and diagnostic criteria were described previously [20]. Participants were followed annually using similar evaluations at each visit. Systematic information on demographics, diagnoses, family history, neuropsychological testing, and medical history were provided by NACC-UDS. Informed consent was given by all participants and co-participants.

### 2.2. Subject Selection/Study Population

Participants who met the National Institute on Aging-Alzheimer’s Association (NIA-AA) or National Institute of Neurological and Communicative Disorders and Stroke/Alzheimer’s Disease and Related Disorders Association (NINCDS/ADRDA) criteria for AD [21,22] were identified. Participants diagnosed with diabetes were included in the study. Participants who did not have either AD or diabetes were excluded. Subjects with type 1 diabetes, diabetes insipidus, gestational diabetes, or latent autoimmune diabetes were excluded. Participants were classified into either the metformin group or the non-metformin group based on their medical history recorded in the UDS.

### 2.3. Measures and Outcomes

Patients with a primary diagnosis of probable AD according to the NINCDS- ADRDA Alzheimer’s criteria [21] at the clinical visit were included in the study. Diabetes was reported based on a clinician assessment of the patient’s medical history at baseline. Information on diabetic treatment was extracted from the UDS Medication Form. Patients receiving metformin therapy were classified as the metformin group. Patients who receive other antidiabetic medications other than metformin were classified as the non-metformin group. Since we did not have information on medication exposure between visits, we assumed a patient was taking a medication over the entire time period if he/she was taking the medication at consecutive visits. Mini-mental state examination (MMSE) was used as the cognitive outcome. Baseline MMSE scores and follow-up MMSE scores were obtained from NACC data. Severe dementia was defined as MMSE score <10 during follow-up [23,24]. The presence of *APOE ε3*/*ε4* or *ε4*/*ε4* genotypes was classified as *APOE ε4* carriers in this study. The *APOE* genotype is run independently by the Alzheimer’s Disease Centers and reported to NACC on the NACC Neuropathology Form. The *APOE* genotype is also reported by the Alzheimer’s Disease Genetics Consortium (ADGC) and the National Institute of Aging Genetics of Alzheimer’s Disease Data Storage Site (NCRAD).

### 2.4. Statistical Analysis

We used *t*-tests to examine the mean differences for continuous variables between metformin and nonusers. Chi-square tests were utilized to examine the distribution differences for categorical variables. Propensity score matching was performed to reduce the selection bias by balancing baseline characteristics: age, gender, education level (years), MMSE score at baseline, comorbidities (hypertension, atrial fibrillation, and congestive heart failure), and medications (dementia medication, antihypertensive medication, and antidepressants. Two cohorts were matched at a ratio of 1:1 by using nearest-neighbor matching via the R package MatchIt [25].

The cumulative probability of severe dementia was compared using the Kaplan–Meier method. Multivariable adjusted Cox proportional hazards regression analyses were performed to test the effect of metformin on the risk of developing severe dementia. A subgroup analysis was performed with participants who were *APOE ε4* carriers or non-carriers. All analyses were performed with R (version 4.1.2, R Foundation for Statistical Computing, Vienna, Austria).

## 3. Results

### 3.1. Baseline Characteristics

From September 2005 to December 2021, 44,713 patients were registered in the NACC. In this study, we included patients with diagnoses of possible or probable AD (n = 21,848). The study inclusion date was defined as the date of AD diagnosis. We excluded patients without a diagnosis of diabetes (n = 19,662) and other types of diabetes (n = 2). Patients were excluded if data were missing on MMSE score at baseline (n = 878) or baseline MMSE score < 10 (n = 113, to avoid floor effects). Finally, we excluded patients who had no follow-up visit (n = 476) and those who had no diabetes medications (n = 402).

A total of 1306 patients with AD and T2DM were eligible, comprising 785 metformin users and 521 non-users (Figure 1). Patients in the non-metformin group were more likely to be older (77.7 vs. 74.0), have a lower MMSE score (23.35 vs. 24.17), and have more comorbid conditions such as hypertension (80.4% vs. 74.5%), atrial fibrillation (12.3% vs. 7.6%), and congestive heart failure (7.9% vs. 4.1%) (Table 1). The metformin group has more *APOE ε4* carriers (40.0% vs. 33.8%) than the non-metformin group. There was no significant difference in education level (14.06 vs. 14.57 years) between the metformin and the non-metformin groups (*p* = 0.30). After propensity score matching, the two cohorts were balanced with no significant difference existing (Table 2).

### 3.2. Severe Dementia Risk

There were 48 subjects (8.6%) in the metformin group and 45 subjects (9.6%) in the non-metformin group who developed severe dementia. During the observation period (2005–2021), the mean time to severe dementia in the metformin group was 147 ± 2.9 months. In the non-metformin group, the mean time to severe dementia was 145 ± 3.4 months.

The cumulative incidence curves for severe dementia stratified by the drug use of metformin after propensity score matching are shown in Figure 2. Kaplan–Meier analyses showed that there were no significant differences in the cumulative probability of remaining severe dementia free between metformin and non-metformin groups in the propensity score-matched cohorts (log-rank *p* = 0.56).

Cox proportional hazards regression analyses showed that there was no statistically significant association between the use of metformin and severe dementia (HR, 0.96; 95% CI, 0.63–1.46; *p* = 0.85) after adjusting for age, gender, education level, *APOE ε4*, comorbidities, and medications (Table 3).

### 3.3. Subgroup Analysis of Subjects with APOE ε4

A subgroup analysis was performed with participants who were *APOE ε4* carriers (n = 366) or non-carriers (n = 676). There were 42 *APOE ε4* carriers (11.5%) who developed severe dementia, while 51 *APOE ε4* non-carriers (7.5%) developed severe dementia during follow-up. Kaplan–Meier analyses demonstrated that those receiving metformin did not have a lower cumulative probability of remaining severe dementia free among *APOE ε4* carriers (log-rank *p* = 0.24, Figure 3a). This trend was further confirmed in the subgroup analysis of outcomes in the *APOE ε4* non-carriers (log-rank *p* = 0.81, Figure 3b).

Cox proportional hazards regression analyses demonstrated that metformin was not associated with a lower risk of developing severe dementia (HR, 0.92; 95% CI, 0.48–1.77; *p* = 0.80) among *APOE ε4* carriers after adjusting for covariates (Table 4). Results for *APOE ε4* non-carriers showed no statistically significant associations between metformin use and severe dementia (HR, 1.13; 95% CI, 0.64–2.01; *p* = 0.67).

## 4. Discussion

The present study found that the use of metformin was not associated with a lower risk of developing severe dementia among AD patients with T2DM. The results were consistent in subgroups that stratified patients by their *APOE ε4* status. Analysis of the subgroup of *APOE ε4* carriers showed that there was no significant difference in time to developing severe dementia between the metformin users and non-users. Similar results were found in the group of *APOE ε4* non-carriers.

It is well received that metformin is associated with a lower risk of dementia, however, its effects on cognitive decline have been inconsistent. Samaras et al. demonstrated that metformin use was associated with a slower cognitive decline over time in cognitively normal elderly people [26]. Their findings were in consistent with a previous review article which indicated that metformin was associated with a decreased risk of incident dementia [14]. Conversely, a longitudinal study by Wennbery et al. reported no statistically significant associations between metformin use and cognitive test performance alteration [27]. While most of the previous studies have investigated associations between metformin and dementia risk in cognitively normal people with diabetes, few studies have evaluated the relationship between metformin use and cognitive outcomes among people with diabetes who also have a diagnosis of AD. The present study distinguishes itself from previous research by including diabetes patients who already have a diagnosis of AD and evaluating the impact of metformin on time to severe dementia. Our results are consistent with a recently published study by Wu et al. [19]. They examined the association between dilatates medications and memory decline over time using mixed-effects models and found no relationship between metformin and memory decline in AD patients. A double-blind placebo-controlled randomized pilot trial [28] also found that there was no significant relationship between metformin and memory change in patients with mild cognitive impairment. Those findings altogether suggested that in the presence of cognitive impairment due to AD, metformin might no longer be neuroprotective.

The *APOE ε4* allele is a key genetic risk factor for cognitive decline, AD, and dementia [29,30]. *APOE ε4* is associated with a higher prevalence of AD and an earlier age of onset [31]. It has been reported that individuals carrying the *APOE ε4* allele experience a more rapid cognitive decline compared to those who are non-carriers of *APOE ε4* [32,33]. Our results were consistent with those findings by observing more cases of severe dementia among *APOE ε4* carriers compared with the *APOE ε4* non-carriers (11.5% vs. 7.5%). Numerous studies have explored the influence of the *APOE* genotype on responses to interventions. Beneficial effects among *APOE ε4* carriers were reported in patients with mild cognitive impairment. Interventions included galantamine [34], docosahexaenoic acid [35], and Mediterranean diet-based interventions [36]. Recent research has spotlighted the role of the *APOE ε4* carrier status as a modifier of metformin’s impact on cognitive decline in AD patients [19]. This suggests that the therapeutic effects of metformin might be modulated by the *APOE ε4* genotype [28]. Patients without the *APOE-ε4* allele treated with metformin might have better cognitive outcomes than those with the *APOE-ε4* allele. Nonetheless, our subgroup analysis revealed that the *APOE ε4* genotype did not significantly modify the relationship between metformin use and incidence of severe dementia. Interpreting the absence of significant interactions requires caution as it could be influenced by the relatively small sample size. Despite having a relatively large cohort and a higher prevalence of the *APOE ε4* allele in the NACC data compared to the general population, there are still limitations in evaluating interactions due to the sample size. Further studies are warranted to elucidate whether *APOE ε4* carriers might potentially derive greater benefits from metformin usage.

To the best of our knowledge, this is the first study evaluating the relationship between metformin use and time to develop severe dementia among people with diabetes who also have a diagnosis of AD. This study has multiple strengths, including a large, population-based sample of AD patients with diabetes, a thorough investigation of potential confounders and effect modifiers, and a longitudinal design. Still, limitations must also be considered. First, although we attempted to address differences in metformin users and non-users through the use of propensity score matching, there may still be unmeasured confounding variables. Second, there was insufficient data on the duration of metformin use and the severity of diabetes, as indicated by factors such as HbA1c levels. These unreported details may introduce residual confounding, as variations in treatment duration and disease severity can significantly impact outcomes. Third, the absence of drug exposure history prior to entry into the database introduces potential bias. It is crucial to recognize that the lack of data on medication use prior to inclusion may influence the interpretation of treatment effects. Additionally, the dataset did not facilitate an assessment of medication adherence, a factor that can influence treatment outcomes. Despite these limitations, the study provides valuable insights into the relationship between metformin use and cognitive outcomes in individuals with AD and diabetes.

## 5. Conclusions

In this large cohort of individuals living with both diabetes and AD, our study did not identify a significant reduction in the risk of developing severe dementia among AD patients with T2DM who were prescribed metformin. A further subgroup analysis, stratified by *APOE ε4* carrier status, revealed that metformin use was not associated with a decreased risk of severe dementia in either *APOE ε4* carriers or non-carriers. Additional larger-scale longitudinal studies are warranted to confirm these findings.

## Figures and Tables

**Figure 1 biomedicines-11-02935-f001:**
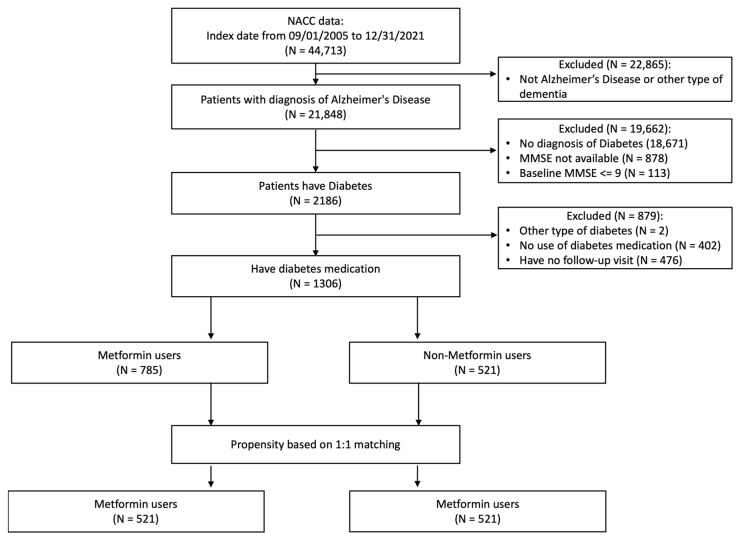
Flowchart of participants included in the study.

**Figure 2 biomedicines-11-02935-f002:**
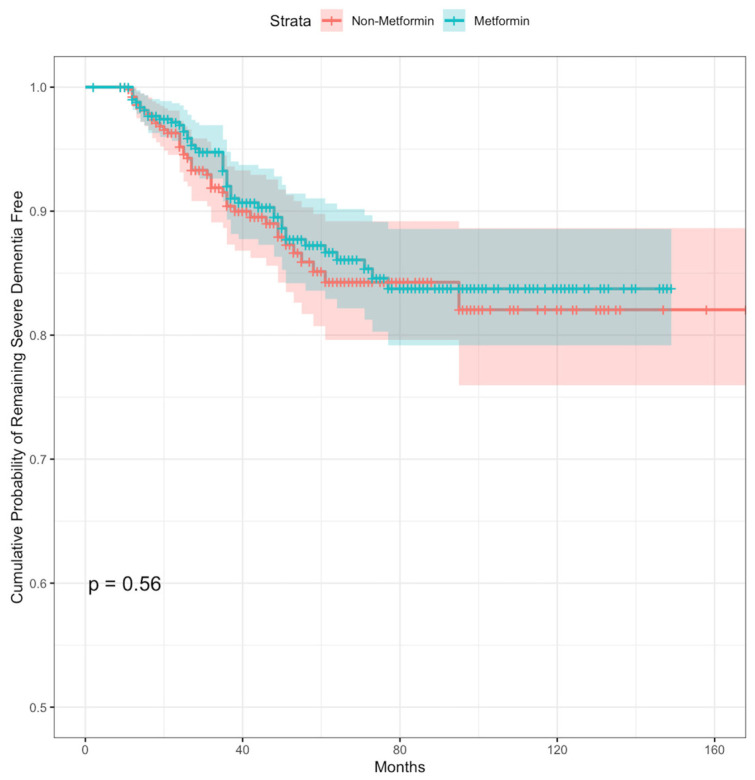
Kaplan-Meier curves examining the cumulative probability of remaining severe dementia free subgroup analysis.

**Figure 3 biomedicines-11-02935-f003:**
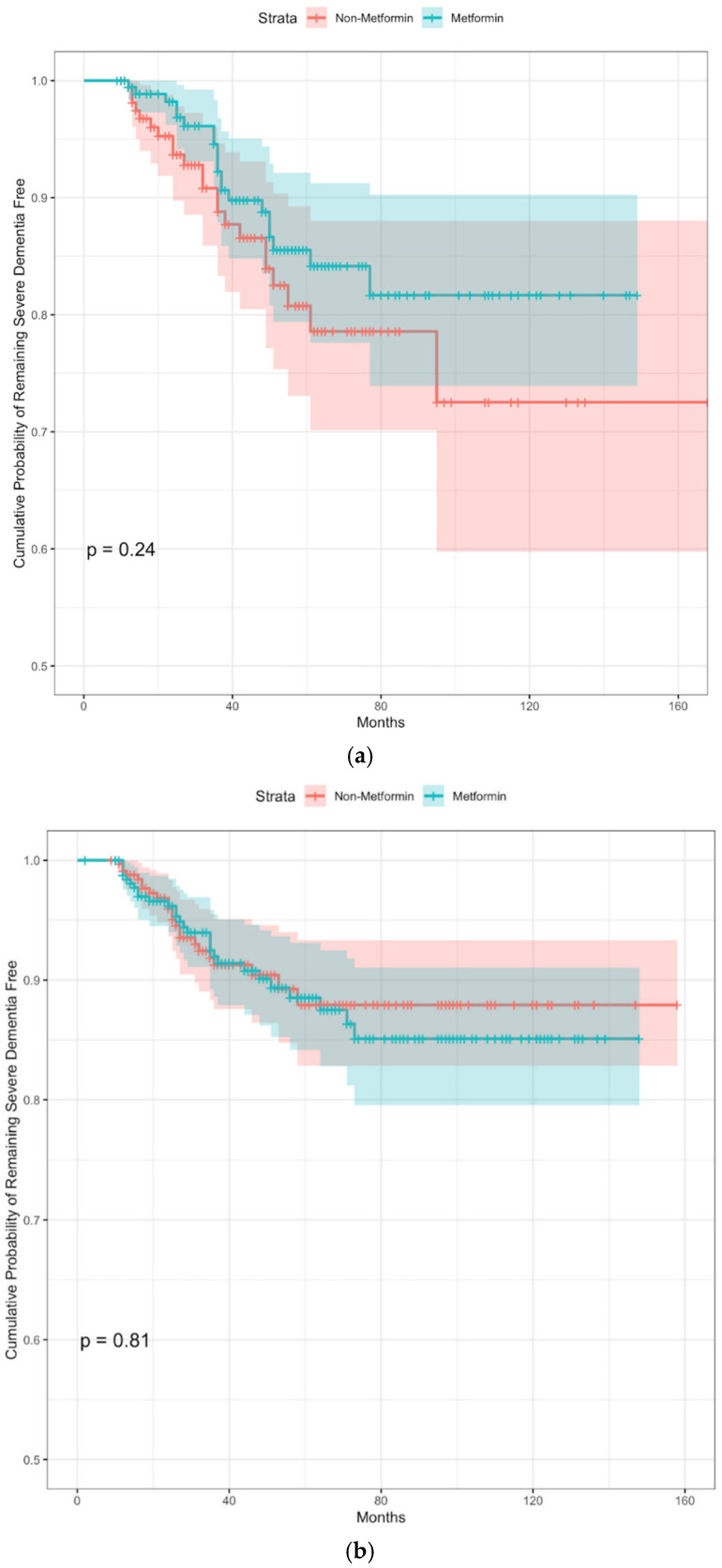
Subgroup analysis of cumulative probability of remaining severe dementia free in *APOE*
*ε4* non-carriers and carriers. (**a**) Subgroup analysis of outcomes in *APOE*
*ε4* carriers; (**b**) Subgroup analysis of outcomes in *APOE*
*ε4* non-carriers.

**Table 1 biomedicines-11-02935-t001:** Characteristics of metformin and non-metformin cohorts (before propensity score matching).

Characteristic	Non-Metformin Cohort N = 521	Metformin Cohort N = 785	*p*-Value
Sex (%)	241 (46.3)	389 (49.6)	0.28
Age (mean (SD))	77.75 (7.51)	73.97 (7.90)	<0.001
Educational level (years)	14.06 (7.51)	14.57 (9.01)	0.29
Baseline MMSE score	23.35 (4.78)	24.17 (4.54)	0.002
*APOE ε4* carrier (%)	176 (33.8)	314 (40.0)	0.027
Hypertension (%)	419 (80.4)	585 (74.5)	0.016
Atrial fibrillation (%)	64 (12.3)	60 (7.6)	0.007
Congestive heart failure (%)	41 (7.9)	32 (4.1)	0.005
Dementia medication (%)	218 (41.8)	321 (40.9)	0.78
Antihypertensive medication (%)	432 (82.9)	605 (77.1)	0.013
Antidepressant (%)	162 (31.1)	278 (35.4)	0.12

**Table 2 biomedicines-11-02935-t002:** Characteristics of metformin and non-metformin cohorts (propensity score-matched).

Characteristic	Non-Metformin Cohort N = 521	Metformin Cohort N = 521	*p*-Value
Sex (%)	241 (46.3)	250 (48.0)	0.62
Age (mean (SD))	77.75 (7.51)	77.04 (6.82)	0.11
Educational level (years)	14.06 (7.51)	14.04 (7.75)	0.55
Baseline MMSE score	23.35 (4.78)	23.80 (4.79)	0.13
*APOE ε4* carrier	176 (33.8)	195 (37.4)	0.24
Hypertension (%)	419 (80.4)	419 (80.4)	1.00
Atrial fibrillation (%)	64 (12.3)	54 (10.4)	0.38
Congestive heart failure (%)	41 (7.9)	30 (5.8)	0.22
Dementia medication (%)	218 (41.8)	208 (39.9)	0.57
Antihypertensive medication (%)	432 (82.9)	420 (80.6)	0.38
Antidepressant (%)	162 (31.1)	171 (32.8)	0.60

**Table 3 biomedicines-11-02935-t003:** Analyses of the association of use of metformin with risk of developing severe dementia.

	HR (95% CI)	*p* Value
**Multivariable Adjusted Analysis ^a^**		
No metformin use	1 [Reference]	1 [Reference]
Metformin use	0.96 (0.63–1.46)	=0.85
**Unadjusted Analysis**		
No metformin use	1 [Reference]	1 [Reference]
Metformin use	0.89 (0.59–1.33)	=0.56

Abbreviations: HR, hazard ratio; CI, confidence interval. ^a^ Adjusted for age; gender; education level; baseline MMSE score; status of *APOE ε4* carrier; diagnosis of hypertension; congestive heart failure; atrial fibrillation; use of dementia medication, antihypertensive medication, and antidepressants.

**Table 4 biomedicines-11-02935-t004:** Cox proportional hazard regression analysis for *APOE ε4* carrier and non-carriers.

	*APOE ε4* Carrier (366)					*APOE ε4* Non-Carrier (676)				
	Users (n = 190)		Non-Users (n = 176)		Users vs. Non-Users	Users (n = 331)		Non-Users (n = 345)		Users vs. Non-Users
	Events	Event Rate	Events	Event Rate	Adjusted HR ^a^ (95%CI)	Events	Event Rate	Events	Event Rate	Adjusted HR ^a^ (95%CI)
Severe dementia	20	10.5%	22	12.5%	0.92 (0.48–1.77), *p* = 0.80	28	8.5%	23	6.7%	1.13 (0.64–2.01), *p* = 0.67

Abbreviations: HR, hazard ratio; CI, confidence interval. ^a^ Adjusted for age; gender; education level; baseline MMSE score; status of *APOE ε4* carrier; diagnosis of hypertension; congestive heart failure; atrial fibrillation; use of dementia medication, antihypertensive medication, and antidepressants.

## Data Availability

Data was obtained from NACC and are available [https://naccdata.org] with the permission of NACC.

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
