# Peer review of "The Association between Metformin Use and Risk of Developing Severe Dementia among AD Patients with Type 2 Diabetes"

_biomedicines, 2023, doi:10.3390/biomedicines11112935_

Round 1

Reviewer 1 Report

the manuscript is interesting and generally well written. Only minor revisions are necessary. In particular:

Lines 56-57: It deserves to be mentioned that metformin also showed important anticancer effects (as recently reviewed PMID: 36361682 )

Tables: statistical significant differences should be written in bold

Differences in early onset AD should also be evaluated 

Genes' names must be written in italic

Author Response

Response to reviewer#1:

Reviewer's Comment 1: Lines 56-57: It deserves to be mentioned that metformin also showed important anticancer effects (as recently reviewed PMID: 36361682)

Response: Thank you for pointing out the potential anticancer effects of metformin. However, this study is mainly focused on investigating the potential benefits of metformin in reducing the risk of developing AD. We will review the reference (PMID: 36361682) thoroughly and take it into account in our future research work discussing metformin’s anticancer effect.

Reviewer's Comment 2: Tables: statistical significant differences should be written in bold

Response: We appreciate your suggestion to highlight statistically significant differences in our tables. In the revised manuscript, we ensured that all statistically significant results are presented in bold text for clarity and emphasis.

Reviewer's Comment 3: Differences in early onset AD should also be evaluated

Response:  Our study distinguishes itself from previous research by including diabetes patients who already have a diagnosis of AD and evaluating the impact of metformin on time to severe dementia.  Thank you for raising the issue of early onset Alzheimer's Disease (AD). We will include cognitive normal patients (different patient group) in the future research to compare the differences in early onset AD among metformin users and non-users.

Reviewer's Comment 4: Genes' names must be written in italic

Response: We appreciate your attention to detail regarding the formatting of gene names. In the revised manuscript, we ensured that gene names are written in italic font to adhere to standard scientific formatting conventions.

Reviewer 2 Report

This is an interesting study. However, some problems could be identified:

-“Diabetes mellitus (DM) is a metabolic disorder characterized by chronic hyperglycemia resulting from a range of mechanisms leading to pancreatic β cell failure” – this is not completely correct for all types of DM

-“Numerous studies have shown that individuals with T2DM often exhibit worse cognitive performance compared to those who do not have diabetes [4-6]. Cognitive impairments, including deficits in memory, attention, and executive function, are more prevalent in T2DM patients. These findings indicate a potential association between T2DM and cognitive decline, which could be relevant to the development of AD.” – some repetition of ideas exist here

-in my opinion, the novelty of this work should be better demonstrated. In the sentence “Few studies have evaluated the relationship between metformin use and cognitive outcomes among people with diabetes who also have a diagnosis of AD.”, authors must include the adequate references of the “few studies”. In addition, other works in this context must also be collected and included in the manuscript (e.g. https://www.mdpi.com/2075-4426/13/5/738 )

-Ethics concerns must be clearly explained, as this work involved individuals with cognitive impairments

-“The presence of APOE ε3/ε4 or ε4/ε4 genotype was classified as APOE ε4 carriers in this study.” Authors must clarify in the manuscript how this information was acquired.

Minor editing of English language required

Author Response

Response to reviewer:

Thank you for your thoughtful feedback on our study. We appreciated your insights. We’ve addressed the points you raised in order to improve the quality and accuracy of our work.

Regarding the statement about diabetes mellitus (DM) being characterized by chronic hyperglycemia and pancreatic β cell failure, we acknowledge that there are different types of DM with varying underlying mechanisms. We revised the description to reflect this nuance more accurately.

We understand that there might be repetition in the section discussing the association between type 2 diabetes mellitus (T2DM) and cognitive decline. We eliminated redundancies and present the information more cohesively.

To enhance the novelty of our work, we will provide appropriate references for the statement about the scarcity of studies evaluating the relationship between metformin use and cognitive outcomes. Additionally, we will incorporate relevant studies, including the one you've mentioned (https://www.mdpi.com/2075-4426/13/5/738), to provide a comprehensive context for our research. This study explores the impact of metformin use on dementia risk among diabetes mellitus (DM) patients, revealing that lower metformin intensity is associated with a reduced risk of dementia. The findings of this study support our idea that “However, most of the previous studies focused on the associations between metformin and dementia risk in cognitive normal people with type 2 diabetes” and “Few studies have evaluated the relationship between metformin use and cognitive outcomes among people with diabetes who also have a diagnosis of AD.” Our study distinguishes itself from previous research by including diabetes patients who already have a diagnosis of AD and evaluating the impact of metformin on time to severe dementia. 

This study has no ethic concerns. The NACC data are de-identified and informed consent is obtained from all participants at the individual ADRCs.

APOE genotype is run independently by the ADC and reported to NACC on the NACC Neuropathology Form. APOE genotype is also reported by Alzheimer’s Disease Genetics Consortium (ADGC) and National Institute of Aging Genetics of Alzheimer’s Disease Data Storage Site (NCRAD). We have added description to clarify.

Round 2

Reviewer 2 Report

Improvements are still necessary:

-"various mechanisms that result in sufficient insulin production" (insufficient?)

-I maintain that the novelty of his work must be better demonstrated in the manuscript

-concerning Ethics, despite authors referred "Recruitment, participant evaluation, and diagnostic criteria were de- 91 scribed previously [19]. Participants were followed annually using similar evaluations at 92 each visit. Systematic information on demographics, diagnoses, family history, neuropsy- 93 chological testing and medical history were provided by NACC-UDS. Informed consent 94 was given by all participants and co-participants.", given the fact that this work involved individuals with cognitive impairments, Ethics concerns must be clearly explained in the manuscrit

Author Response

Comment#1: "various mechanisms that result in sufficient insulin production" (insufficient?)

Response: Thanks for pointing it out. It should be insufficient. The sentence has been revised accordingly.

Comment#2:

-I maintain that the novelty of his work must be better demonstrated in the manuscript

-concerning Ethics, despite authors referred "Recruitment, participant evaluation, and diagnostic criteria were de- 91 scribed previously [19]. Participants were followed annually using similar evaluations at 92 each visit. Systematic information on demographics, diagnoses, family history, neuropsy- 93 chological testing and medical history were provided by NACC-UDS. Informed consent 94 was given by all participants and co-participants.", given the fact that this work involved individuals with cognitive impairments, Ethics concerns must be clearly explained in the manuscript

Response:

Thanks for the suggestion. This study has no ethic concerns. The NACC data are de-identified and informed consent is obtained from all participants at the individual ADRCs. We have added a paragraph in the manuscript to explain.

“The data underwent de-identification procedures and adhered to the patient confidentiality standards outlined in the Health Insurance Portability and Accountability Act (HIPAA). Consequently, approval from an ethics committee was not required.”

Round 3

Reviewer 2 Report

The document was improved. However, authors referred that "Regrettably, few studies have evaluated the relationship between metformin use and cognitive outcomes among people with diabetes who also have a diagnosis of AD" - these studies MUST be referenced in the manuscript as well as their main results and, thus, the nolevty of the present work.

Author Response

Thanks for the comments. In this sentence, we're conveying that there is a scarcity of research on the specific topic (no similar study was found). To enhance the novelty of our work, we provided appropriate references for the statement about the scarcity of studies evaluating the relationship between metformin use and cognitive outcomes. Besides the reference you've mentioned (https://www.mdpi.com/2075-4426/13/5/738), we have added another reference (https://bmjopen.bmj.com/content/bmjopen/9/7/e024954.full.pdf ). This study evaluates the association between metformin treatment and the risk of neurodegenerative disease (ND) among elderly adults with type 2 diabetes by using the Veterans Affairs (VA) database (2004–2010). The authors suggest that metformin treatment may be associated with a reduced risk of neurodegenerative disease among elderly adults with type 2 diabetes mellitus, however, this study excluded those with neurodegenerative disease at baseline and only focused on cognitive normal patients. The findings of this study also support our statement that “However, most of the previous studies focused on the associations between metformin and dementia risk in cognitive normal people with type 2 diabetes” and “Few studies have evaluated the relationship between metformin use and cognitive outcomes among people with diabetes who also have a diagnosis of AD.” Our study distinguishes itself from previous research by including diabetes patients who already have a diagnosis of AD and evaluating the impact of metformin on time to severe dementia. 

We added another sentence in the last paragraph of discussion section: “To the best of our knowledge, this is the first study evaluating the relationship between metformin use and developing severe dementia among people with diabetes who also have a diagnosis of AD.”